# Blood Biomarkers to Predict Long-Term Mortality after Ischemic Stroke

**DOI:** 10.3390/life11020135

**Published:** 2021-02-10

**Authors:** Laura Ramiro, Laura Abraira, Manuel Quintana, Paula García-Rodríguez, Estevo Santamarina, Jose Álvarez-Sabín, Josep Zaragoza, María Hernández-Pérez, Xavier Ustrell, Blanca Lara, Mikel Terceño, Alejandro Bustamante, Joan Montaner

**Affiliations:** 1Neurovascular Research Laboratory, Vall d’Hebron Institute of Research (VHIR), Universitat Autònoma de Barcelona, 08035 Barcelona, Spain; laura.ramiro@vhir.org (L.R.); paula.garcia.rodriguez@vhir.org (P.G.-R.); joan.montaner@vhir.org (J.M.); 2Epilepsy Unit, Neurology Department, Vall d’Hebron University Hospital, Universitat Autònoma de Barcelona, 08035 Barcelona, Spain; laura.abraira@vhir.org (L.A.); maquintavh@gmail.com (M.Q.); esantama@vhebron.net (E.S.); 3Department of Medicine, Universitat Autònoma de Barcelona, 08193 Barcelona, Spain; josalvarez@vhebron.net; 4Neurology Department, Vall d’Hebron University Hospital, Universitat Autònoma de Barcelona, 08035 Barcelona, Spain; 5Stroke Unit, Verge de la Cinta University Hospital, 43500 Tortosa, Spain; jzaragoza.ebre.ics@gencat.cat; 6Stroke Unit, Germans Trias i Pujol University Hospital, 08916 Badalona, Spain; mhernandez@igtp.cat; 7Stroke Unit, Joan XXIII University Hospital, 43005 Tarragona, Spain; xustrell.hj23.ics@gencat.cat; 8Stroke Unit, Bellvitge University Hospital, 08907 Barcelona, Spain; blara@bellvitgehospital.cat; 9Stroke Unit, Josep Trueta University Hospital, 17007 Girona, Spain; mterceno.girona.ics@gencat.cat; 10Stroke Research Program, Institute of Biomedicine of Seville, IBiS/Hospital Universitario Virgen del Rocío/CSIC/University of Seville & Department of Neurology, Hospital Universitario Virgen Macarena, 41009 Seville, Spain

**Keywords:** ischemic stroke, biomarker, mortality, endostatin, IL-6, TNF-R1

## Abstract

Stroke is a major cause of disability and death globally, and prediction of mortality represents a crucial challenge. We aimed to identify blood biomarkers measured during acute ischemic stroke that could predict long-term mortality. Nine hundred and forty-one ischemic stroke patients were prospectively recruited in the Stroke-Chip study. Post-stroke mortality was evaluated during a median 4.8-year follow-up. A 14-biomarker panel was analyzed by immunoassays in blood samples obtained at hospital admission. Biomarkers were normalized and standardized using *Z*-scores. Multiple Cox regression models were used to identify clinical variables and biomarkers independently associated with long-term mortality and mortality due to stroke. In the multivariate analysis, the independent predictors of long-term mortality were age, female sex, hypertension, glycemia, and baseline National Institutes of Health Stroke Scale (NIHSS) score. Independent blood biomarkers predictive of long-term mortality were endostatin > quartile 2, tumor necrosis factor receptor-1 (TNF-R1) > quartile 2, and interleukin (IL)-6 > quartile 2. The risk of mortality when these three biomarkers were combined increased up to 69%. The addition of the biomarkers to clinical predictors improved the discrimination (integrative discriminative improvement (IDI) 0.022 (0.007–0.048), *p* < 0.001). Moreover, endostatin > quartile 3 was an independent predictor of mortality due to stroke. Altogether, endostatin, TNF-R1, and IL-6 circulating levels may aid in long-term mortality prediction after stroke.

## 1. Introduction

Stroke is a leading cause of death and long-term disability worldwide. In fact, in 2017, someone died due to stroke every 3.5 min, a total of 2.7 million individuals in the whole year, so it is a major health and socioeconomic concern [1]. An early prediction of this fatal outcome after cerebral ischemia can facilitate decision-making processes, such as patients’ inclusion to clinical trials or early admission to specialized stroke units, ultimately optimizing patients’ management. However, predicting stroke outcome remains challenging. Various attempts to identify characteristics associated with poor outcomes have been made. Different predictive models for mortality in stroke patients have been developed, with an accuracy around 80% in most models [2,3,4,5], although none of them are routinely used in clinical practice yet. Many of these models include clinical variables, such as age and stroke severity, as predictors of poor outcome and mortality [6,7,8]. It has been suggested that the addition of blood biomarkers to a clinical model might improve its prediction capacity. However, the vast majority of studies have explored the role of blood biomarkers in predicting outcome and mortality early after stroke, being the outcome measured in most studies 3 months after the event [9,10,11,12]. Whether circulating proteins measured acutely after ischemic stroke can predict long-term mortality is still unclear.

For this reason, in the present study, we aimed to analyze whether blood biomarkers measured during acute ischemic stroke could predict its long-term mortality. To that end, the association between 14 molecules analyzed in the blood in the acute phase of stroke and five-year mortality was assessed in 941 ischemic stroke patients.

## 2. Materials and Methods

### 2.1. Study Design

A prospective longitudinal study was conducted to evaluate long-term mortality in 941 adult patients with ischemic stroke, previously recruited in the Stroke-Chip study (PR(AG)80/2012) [13]. In summary, inclusion criteria were suspected stroke at first medical assessment, with persistent symptoms when arriving at the emergency room; age > 18 years; <6 h from symptom onset to blood sample collection; blood collection preceding thrombolytic treatment; and signed informed consent. The only exclusion criteria was impossibility of collecting blood samples. Moreover, patients with an unclear diagnosis 1 month after the index event were excluded from the analysis. Stroke diagnosis was performed by trained neurologists at each center, according to the World Health Organization [14] definition, and confirmed by neuroimaging. Clinical and demographic data, as well as biomarker results obtained during the acute phase in the Stroke-Chip study, were analyzed with data obtained over a median follow-up of 4.8 years to explore predictors of long-term mortality.

The study was approved by the Ethics Committee of each recruiting center, and all patients or relatives gave written informed consent.

### 2.2. Blood Sample Collection and Biomarker Measurement

Blood samples were drawn at hospital admission, within 6 h after symptom onset, and before any treatment was given. Blood was collected into ethylenediaminetetraacetic acid (EDTA) tubes, centrifuged at 1500× *g* for 15 min at 4 °C, and plasma aliquots were frozen at −80 °C until biomarker analysis. A 14-biomarker panel was used in the original study to distinguish between strokes and stroke mimics, as well as ischemic stroke from intracranial hemorrhage, and these were analyzed in the present study. This panel included apolipoprotein CIII (ApoC-III), D-dimer, endostatin, Fas ligand (FasL), growth-related oncogene-α (GROA), heat shock 70 kDa protein-8 (Hsc70), insulin-like growth factor-binding protein-3 (IGFBP-3), interleukin-6 (IL-6), neuron cell adhesion molecule (NCAM), N-terminal pro-B-type natriuretic peptide (NT-proBNP), S100 calcium-binding protein (S100B), tumor necrosis factor receptor-1 (TNF-R1), vascular adhesion protein-1 (VAP-1), and Von Willebrand factor (vWF). Biomarker measurements were performed using immunoassays, according to the manufacturers’ instructions and blinded to clinical diagnoses, as described before [13]. All samples were tested in duplicate, and the mean coefficient of variation (CV) was <20%. Inter-assay variation was checked by testing in duplicate a commercial internal control (human serum type AB, male, from clotted; Sigma-Aldrich, St. Louis, MO, United States, cat #H6914) in every plate. Biomarker values were log-transformed with a base of 10 and divided by the internal control value of each plate. Due to the high intra-assay variability for some molecules, all values were standardized by plate by *Z*-scores (mean 2, standard deviation [SD] 1).

### 2.3. Clinical Assessment

All of the patients included in the present study were contacted by telephone and interviewed, using a structured questionnaire designed precisely for this study. When a patient was not available, a family relative or caregiver was interviewed, and if these attempts failed, data were compiled by chart review (PR(AG)397/2016). Cause of death was recorded for each patient. Mortality due to stroke was considered when the patient died due to the index stroke or a recurrent cerebrovascular event during follow-up.

Clinical and radiological variables related to stroke onset were collected for the original Stroke-Chip study [13]. Stroke-related variables comprise, among others, stroke severity, according to the National Institutes of Health Stroke Scale (NIHSS); symptomatic hemorrhagic transformation, based on the European Cooperative Acute Stroke Study III criteria (any hemorrhagic transformation with worsening of ≥4 NIHSS points); etiology (Trial of ORG 10172 in Acute Stroke Treatment (TOAST) classification) [15]; and affected cerebral artery territory, according to the Oxford Community Stroke Project (OCSP) classification [16].

### 2.4. Statistical Analysis

Statistical analyses were conducted with IBM SPSS Statistics, version 22.0 for Windows (SPSS Inc., Chicago, IL, United States) and R software (version 3.4.4). Mortality rates during follow-up were assessed with the Kaplan–Meier product limit survival method, using the log–rank test to check statistical significance between groups and conducting simple Cox proportional hazard models to determine differences in continuous variables. The “survivalROC” R package was utilized to execute time-dependent receiver-operating characteristic (ROC) curve analyses to calculate quartile cut-offs for biomarkers with *p*-values <0.1 in the simple Cox models with the best specificity and sensitivity to predict mortality; optimal quartile cut-offs were obtained using the Youden index maximum value (sensitivity + specificity − 1). All variables with a *p*-value <0.1 on univariate analysis were entered into multiple Cox regression models with the forward stepwise method, in order to detect factors independently associated with mortality during follow-up. Two models were performed, one with only clinical variables and another with both categorized biomarkers and clinical variables. The results are presented as hazard ratios (HRs) with 95% confidence interval (CIs). ROC curves were obtained to assess the performance of the models and determine their ability to predict mortality. In order to evaluate the additional value to the logistic regression models of the selected biomarker combinations with the clinical predictors, DeLong’s method was used to compare areas under the curve (AUCs) of ROC curves. The integrative discriminative improvement (IDI) and the net reclassification improvement (NRI) were used to evaluate the incremental effect of adding significant biomarkers to the model built only with clinical data. The “survIDINRI” R package was utilized to obtain IDI and NRI indices in prediction models with censored survival data. A *p*-value lower than 0.05 was considered statistically significant in all tests.

## 3. Results

### 3.1. Demographics

Of the 941 ischemic stroke patients in the original Stroke-Chip study [13], four were lost to follow-up, leaving 937 patients that fulfilled the inclusion criteria for the present study.

Baseline demographic and clinical characteristics of stroke patients are shown in Table 1. Mean ± SD age was 72.8 ± 13.0 years, and 53.9% of patients were men. Arterial hypertension was the most common vascular risk factor in the sample (*n* = 686, 73.2%), followed by dyslipidemia (*n* = 461, 49.2%). The median baseline NIHSS score was 7 (interquartile range (IQR) = 3–15), and median follow-up was 4.8 years (IQR = 1.6–5.2) (Table 1).

### 3.2. Clinical Characteristics and Mortality

Three hundred and sixty-three patients (38.7%) died during the 4.8-year follow-up. Table 1 shows demographic and clinical differences between those patients who died and those who survived. Deceased patients were older, had higher prevalence of hypertension and atrial fibrillation, and a higher baseline NIHSS score. Patients’ cause of death during follow-up is summarized in Figure 1, which shows that the index stroke was the most common cause of death among patients (*n* = 107).

In the multiple regression analysis, the independent predictors of mortality were age (hazard ratio (HR) = 1.063, 95% confidence interval (CI) = 1.049–1.078, *p* < 0.001), female sex (HR = 1.559, 95% CI = 1.243–1.956, *p* < 0.001), hypertension (HR = 1.359, 95% CI = 1.028–1.797, *p* = 0.031), previous modified Rankin scale (mRS) (HR = 1.247, 95% CI = 1.152–1.351, *p* < 0.001), glycemia (HR = 1.002, 95% CI = 1.000–1.004, *p* = 0.043) and baseline NIHSS (HR = 1.089, 95% CI = 1.089–1.073, *p* < 0.001).

### 3.3. Blood Biomarkers and Mortality

Patients who died during the follow-up time had higher baseline levels of D-dimer, endostatin, IL-6, NT-proBNP, VAP-1, vWF, and TNF-R1, and lower levels of apolipoprotein CIII (APOC-III) than patients who survived. For APOC-III, D-dimer, endostatin, IL-6, NT-proBNP, vWF, and TNF-R1, the optimal quartile cut-off point was quartile 2, whereas for VAP-1, quartile 3 was optimal (Table 2).

In the multiple regression model, after adjusting for those clinical variables associated with mortality in univariate analysis, endostatin > quartile 2 (HR = 1.373, 95% CI = 1.061–1.776, *p* = 0.016), TNF-R1 > quartile 2 (HR = 1.392, 95% CI = 1.071–1.808, *p* = 0.013) and IL-6 > quartile 2 (HR = 1.316, 95% CI = 1.025–1.690, *p* = 0.032) were independent predictors of long-term mortality, together with age (HR = 1.058, 95% CI = 1.042–1.073, *p* < 0.001), female sex (HR = 1.674, 95% CI = 1.308–2.142, *p* < 0.001), hypertension (HR = 1.338, 95% CI = 0.992–1.806, *p* = 0.057), previous mRS (HR = 1.232, 95% CI = 1.131–1.343, *p* < 0.001), glycemia (HR = 1.002, 95% CI = 1.000–1.004, *p* = 0.071) and baseline NIHSS (HR = 1.087, 95% CI = 1.069–1.105, *p* < 0.001).

The risk of death increased up to 69% when these three biomarkers were combined (Figure 2). The area under the ROC curve (AUC) of the predictive model was significantly higher when combining biomarkers with clinical variables (85.5%; 95% CI = 81.7–87.5%) than when clinical variables were used alone (84.4%; 95% CI = 82.9–88.1%) (De Long’s test *p* = 0.017) (Figure 3).

The IDI value was 0.022 (95% CI = 0.007–0.048, *p* < 0.001) (IDI events = 0.011 and IDI non-events = −0.011), and the NRI was 0.218 (95% CI = 0.157–0.349, *p* < 0.001) (NRI events = 0.600 and NRI non-events = 0.382), so the inclusion of the biomarkers significantly improved the model when compared to the model built only with clinical variables.

### 3.4. Mortality Due to Stroke

One hundred and thirty-three patients (14.2%) died due to stroke during follow-up. Of those, 107 (29.5%) died due to the index event, and 26 (7.2%) due to a recurrent stroke. Table 1 shows demographic and clinical differences between those patients who died because of stroke and those who survived. Patients who died due to stroke were older, had higher prevalence of hypertension, atrial fibrillation, and coronary artery disease, and had a higher baseline NIHSS score.

In the multivariate analysis, the independent predictors of mortality due to stroke were age (HR = 1.058, 95% CI = 1.037–1.079, *p* < 0.001), hypertension (HR = 1.852, 95% CI = 1.131–3.030, *p* = 0.014) and basal NIHSS (HR = 1.138, 95% CI = 1.111–1.167, *p* < 0.001).

Patients who died because of stroke during the follow-up time had higher levels of D-dimer, endostatin, IL-6, NT-proBNP, VAP-1, vWF, and TNF-R1 than patients who did not die. Each biomarker was divided into quartiles and optimal quartile cut-off points to predict long-term mortality due to stroke. For D-dimer, IL-6, NT-proBNP, vWF, and TNF-R1, the optimal quartile cut-off point was quartile 2, whereas for endostatin and VAP-1 quartile 3 was the optimal (Table 3).

In the multiple regression model, after adjusting for significant clinical variables, endostatin > quartile 3 was an independent predictor of mortality due to stroke (HR = 1.835, 95% CI = 1.196–2.815, *p* = 0.005), together with age (HR = 1.054, 95% CI = 1.029–1.079, *p* < 0.001), hypertension (HR = 1.626, 95% CI = 0.953–2.772, *p* = 0.074), and basal NIHSS (HR = 1.144, 95% CI = 1.110–1.178, *p* < 0.001). The AUC of the predictive model was moderately higher when combining biomarkers with clinical variables (83.6%; 95% CI = 80.2–86.9%) than when clinical variables were used alone (82.5%; 95% CI = 79.1–86.0%), although this difference was not significant (De Long’s tests *p* = 0.066) (Figure 4). The inclusion of the biomarkers improved the model when compared to the model built only with clinical variables, given that the IDI value was 0.020 (95% CI = 0.002–0.050, *p* = 0.02) (IDI events = 0.016 and IDI non-events = −0.004) and the NRI was 0.296 (95% CI = 0.043–0.390, *p* = 0.04) (NRI events = 0.472 and NRI non-events = 0.176).

## 4. Discussion

The present study explored the association of 14 blood biomarkers measured in the acute phase with long-term mortality after ischemic stroke. We followed up with a cohort of 941 ischemic stroke patients over a median time of almost 5 years after the event, in order to determine whether some blood biomarkers can predict mortality in this population. The proportion of patients that die after ischemic stroke is influenced by different variables, such as subtype of stroke or health resources of each country. According to the literature, 40–60% of ischemic stroke patients die within 5 years after the event [17,18,19]. In the present study, we found that 38.7% of ischemic stroke patients died during the median 4.8-year follow-up period, and those who died were older, had higher prevalence of some vascular risk factors, such as hypertension and atrial fibrillation, and had a higher baseline NIHSS score.

The independent predictors of mortality were age, sex, hypertension, previous mRS, and basal NIHSS, which are among the most frequent clinical and demographical variables associated with stroke mortality, as reported in the literature [8,20]. In addition, from the proposed biomarkers, we found that after adjusting for clinical variables, increased levels of endostatin, IL-6, and TNF-R1 were independent predictors of long-term mortality following stroke. In fact, when these three biomarkers were combined, the risk of death increased up to 69%. The best biomarkers identified in the present study are known key players in pathophysiological pathways implicated in stroke, such as angiogenesis (endostatin) and inflammatory response (IL-6 and TNF-R1).

It is widely reported that cerebral ischemia triggers an inflammatory response both in the brain and peripheral circulation, which leads to up-regulation of inflammatory cytokines such as IL-6 [21]. Increased circulating levels of IL-6 have been regularly associated with poor-outcome after stroke [22]. The value of IL-6 to predict mortality after cerebral ischemia has also been studied before. Various studies support that increased baseline levels of IL-6 are associated with risk of death 3 months after the event [23,24,25,26]. Regarding long-term mortality, it has been revealed that IL-6 could also predict mortality both 1 year [27] and 2 years [28] after stroke. In the present study, we propose that IL-6 could also predict mortality even at later time-points, being associated with increased risk of mortality 5 years after cerebral ischemia.

TNF-R1 is a transmembrane death receptor that is able to detect the presence of extracellular death signals and ultimately trigger cell apoptosis [29]. TNF-R1 can be activated by both membrane-bound and soluble forms of TNF-α [30]. It is well known that within the first hours after cerebral ischemia, cells of the ischemic core and penumbra suffer irreversible damage that ultimately leads to cell death by apoptosis [21]. Increased levels of TNF-α have previously been associated with poor outcome after ischemic stroke [31,32]. However, the association of TNF-R1 in stroke outcome and long-term mortality has never been described before. Here we found that those patients with increased risk of long-term death after stroke showed higher baseline levels of TNF-R1, reinforcing the crucial role of the TNF signaling pathway in stroke prognosis.

Endostatin is derived from type XVIII collagen. It inhibits both the proliferation and migration of endothelial cells, ultimately leading to inhibition of angiogenesis [33]. Regarding outcome, it has been previously reported that tissue plasminogen activator (tPA)-treated stroke patients showing higher endostatin level at hospital admission had an impaired functional outcome three months after the event [34]. In relation to mortality, one study including 3463 acute ischemic stroke patients revealed that increased baseline endostatin levels were associated with increased risk of mortality and severe disability at 3 months [35]. However, to date, the association of endostatin and long-term mortality after stroke has not been explored. Interestingly, in the present study, we found that those patients who did not survive during follow-up had higher baseline levels of endostatin at hospital admission. This is in line with what has previously been reported, and therefore endostatin may be a useful biomarker to predict both early and long-term mortality. In addition to this, we have reported for the first time that baseline endostatin levels are also associated with long-term mortality due to stroke, making endostatin even more interesting as a promising stroke biomarker.

Remarkably, the risk of long-term death when these three biomarkers were combined increased up to 69%, while the risk was below 10% when none of these biomarkers were pathological. The combination of blood biomarkers and clinical variables showed a slight but significant additional predictive value over clinical data in AUC results, although the clinical model included stroke severity. Moreover, these findings are supported by positive results in NRI and IDI procedures. In addition, endostatin was also able to predict long-term mortality due to stroke, given that the combination of endostatin and clinical variables moderately increased the predictive value when compared to clinical data. All in all, Il-6 TNF-R1 and endostatin seem to be interesting candidates for further exploration as long-term mortality biomarkers after stroke. In this regard, the modulation of these molecules and pathways through angiogenic or anti-inflammatory drugs might be an attractive approach to improve patients’ outcome and reduce mortality. In fact, some pre-clinical studies already point in that direction [21].

The main strengths of the present study are the large sample size and the standardized follow-up after a long time of almost five years. However, our study has some limitations. First, in some cases, data were collected by chart review, which might lead to some missing data. Second, the specific cause of death of 20% of patients is unknown, so some deaths due to stroke may not have been diagnosed. Third, the need to standardize the results could have led to an underestimation of the biomarker effect. Finally, these patients were recruited in 2012–2013, so the actual high rates of thrombectomies that are being performed in hospitals, which may contribute to the reduction in mortality, might not be fully represented by the present cohort. For that reason, our results should be replicated in an independent cohort before solid conclusions can be drawn.

In conclusion, we have identified that the acute upregulation of endostatin, IL-6, and TNF-R1 after ischemic stroke can predict long-term mortality. Along this line, a panel comprising these three biomarkers could be used to help identify high-risk patients, so that more aggressive therapeutic strategies can be targeted to those most likely to benefit. Additional studies are needed to validate these findings, in order to elucidate the real impact on stroke patients’ management.

## Figures and Tables

**Figure 1 life-11-00135-f001:**
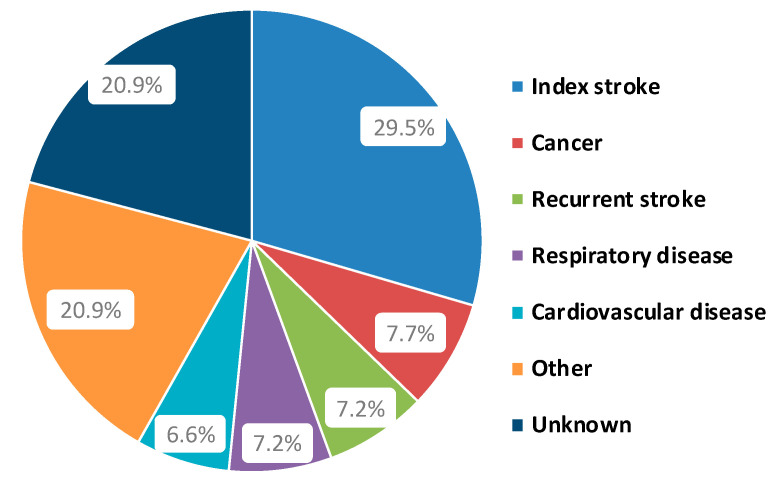
Patients’ cause of death during follow-up.

**Figure 2 life-11-00135-f002:**
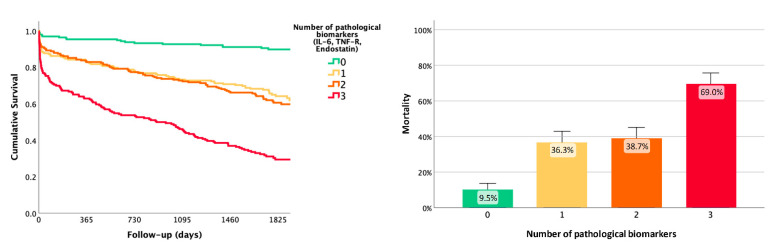
Combination of predictive blood biomarkers (endostatin > quartile 2, TNF-R1 > quartile 2, and IL-6 > quartile 2). Risk of death increased up to 36.3% when adding any of the biomarkers, up to 38.7% when combining two of three biomarkers, and up to 69% when combining all three biomarkers. IL-6: interleukin-6; TNF-R: tumor necrosis factor receptor-1.

**Figure 3 life-11-00135-f003:**
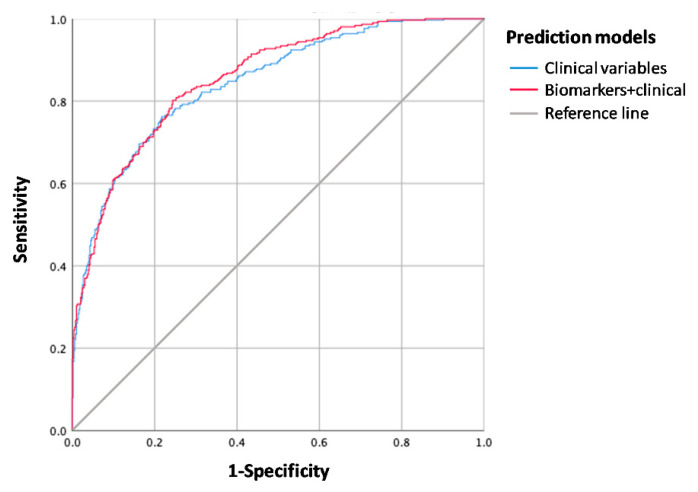
Receiver-operating characteristic (ROC) curve of the predictive model for long-term mortality. The predictive capacity of the model was slightly higher when combining clinical variables and blood biomarkers (endostatin, IL-6, and TNF-R1) than when using clinical variables alone.

**Figure 4 life-11-00135-f004:**
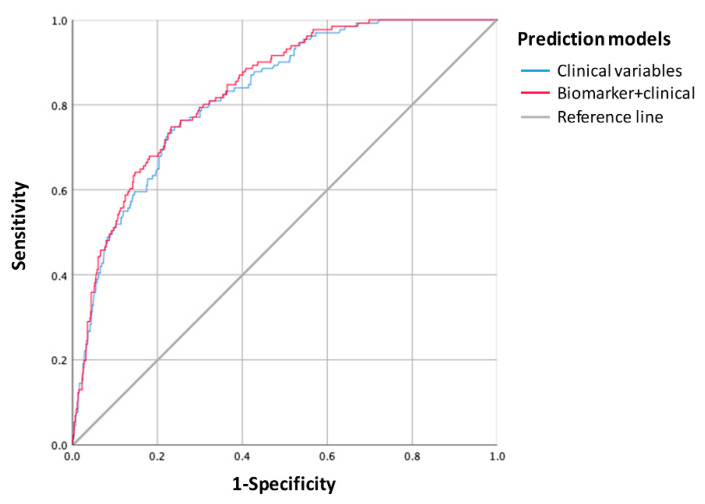
Receiver-operating characteristic (ROC) curve of the predictive model for long-term mortality due to stroke. The predictive capacity of the model was slightly higher when combining clinical variables with a blood biomarker (endostatin) than when using clinical variables alone.

**Table 1 life-11-00135-t001:** Demographic and clinical characteristics of patients.

		*n* = 937	Death during Follow-Up		*p*-Value	Death Due to Stroke		*p*-Value
			No (*n* = 574)	Yes (*n* = 363)		No (*n* = 804)	Yes (*n* = 133)	
Age (years, mean ± SD)		72.8 ± 13.0	68.4 ±12.9	79.7 ± 9.6	<0.001	71.5 ±13.1	80.4 ± 9.7	<0.001
Gender (male, *n*, %)		505 (53.9)	325 (56.6)	180 (49.6)	0.033	435 (54.1)	70 (52.6)	0.683
Smoking (*n*, %)		151 (16.1)	116 (20.2)	35 (9.6)	<0.001	133 (16.5)	18 (13.5)	0.317
Alcoholism (*n*, %)		64 (6.8)	48 (8.4)	16 (4.4)	0.021	55 (6.8)	9 (6.8)	0.828
Arterial Hypertension (*n*, %)		686 (73.2)	391 (68.1)	295 (81.3)	<0.001	573 (71.3)	113 (85.0)	0.001
Diabetes mellitus (*n*, %)		240 (25.6)	121 (21.1)	119 (32.8)	<0.001	199 (24.8)	41 (30.8)	0.107
Dyslipidemia (*n*, %)		461 (49.2)	285 (49.7)	176 (48.5)	0.719	392 (48.8)	69 (51.9)	0.508
Atrial fibrillation (*n*, %)		329 (35.1)	155 (27.0)	174 (47.9)	<0.001	263 (32.7)	66 (59.6)	<0.001
Coronary artery disease (*n*, %)		152 (16.2)	74 (12.9)	78 (21.5)	0.001	121 (15.0)	31 (23.3)	0.017
Previous Stroke (*n*, %)		162 (17.3)	81 (14.1)	81 (22.3)	0.002	132 (16.4)	30 (22.6)	0.086
Previous mRS (median, IQR)		0 (0–1)	0 (0–0)	1 (0–3)	<0.001	0 (0–0)	1 (0–3)	<0.001
Functionality (mRS > 2) (*n*, %)		141 (15.5)	30 (5.4)	111 (31.4)	<0.001	107 (13.7)	34 (26.6)	<0.001
Systolic blood pressure (mmHg)		156.2 ± 28.9	157.3 ± 28.5	154.4 ± 29.5	0.151	156.2 ± 28.9	156.3 ± 28.8	0.993
Diastolic blood pressure (mmHg)		83.0 ± 16.9	84.3 ± 16.8	80.8 ± 16.8	0.006	82.8 ± 16.6	84.0 ± 18.4	0.504
Glycemia (mg/dL)		133.5 ± 46.6	127.5 ± 42.3	142.9 ± 51.3	<0.001	131.6 ± 45.7	144.8 ± 50.1	0.001
Baseline NIHSS score (median [IQR])		7.0 (3–15)	5.0 (2–10)	12.5 (6–19)	<0.001	6.0 (2–12)	17.5 (11–21)	<0.001
Type of ischemic stroke	TIA (*n*, %)	102 (10.9)	80 (13.9)	22 (6.1)	<0.001	100 (12.5)	2 (1.5)	<0.001
	Ischemic stroke (*n*, %)	835 (89.1)	494 (86.1)	341 (93.9)		704 (87.6)	131 (98.5)	
OCSP classification	TACI (*n*, %)	306 (37.6)	123 (25.3)	183 (56.1)	<0.001	214 (31.1)	92 (73.6)	<0.001
	PACI (*n*, %)	310 (38.1)	210 (43.1)	100 (30.7)		288 (41.9)	22 (17.6)	
	LACI (*n*, %)	127 (15.6)	103 (21.1)	24 (7.4)		123 (17.9)	4 (3.2)	
	POCI (*n*, %)	70 (8.6)	51 (10.5)	19 (5.8)		63 (9.2)	7 (5.6)	
TOAST classificaiton	Cardioembolic (*n*, %)	387 (41.6)	210 (36.8)	177 (49.2)	<0.001	325 (40.7)	62 (47.0)	<0.001
	Atherothrombotic (*n*, %)	131 (14.1)	87 (15.2)	44 (12.2)		114 (14.3)	17 (12.9)	
	Lacunar (*n*, %)	120 (12.9)	100 (17.5)	20 (5.6)		118 (14.8)	2 (1.5)	
	Undetermined (*n*, %)	275 (29.5)	158 (27.7)	117 (32.5)		224 (28.0)	51 (38.6)	
	Other causes (*n*, %)	18 (1.9)	16 (2.8)	2 (0.6)		18 (2.3)	0 (0.0)	
Reperfusion therapy (*n*, %)		389 (41.6)	232 (40.5)	157 (43.4)	0.119	322 (40.1)	67 (50.4)	0.017
tPA (*n*, %)		363 (38.8)	213 (37.2)	150 (41.4)	0.037	298 (37.2)	65 (48.9)	0.005
Thrombectomy (*n*, %)		72 (7.7)	46 (8.0)	26 (7.2)	0.770	60 (7.5)	12 (9.0)	0.559
Symptomatic hemorrhagic transformation (*n*, %)		17 (1.8)	1 (0.2)	16 (4.4)	<0.001	4 (0.5)	13 (9.8)	<0.001

mRS: modified Rankin scale; NIHSS: National Institutes of Health Stroke Scale; IQR: interquartile range; TIA: transient ischemic attack; OCSP: Oxfordshire Community Stroke Project; TACI: total anterior circulation infarct; PACI: partial anterior circulation infarct; LACI: lacunar infarct; POCI: posterior circulation infarct; TOAST: Trial of ORG 10172 in Acute Stroke Treatment; tPA: tissue plasminogen activator.

**Table 2 life-11-00135-t002:** Blood biomarker mean values and best quartile cut-off points to predict mortality during follow-up.

Biomarkers	Death during Follow-Up		*p*-Value	Cut-Off (Quartile)	Death during Follow-Up		*p*-Value
	No (*n* = 574)	Yes (*n* = 363)			No (*n* = 574)	Yes (*n* = 363)	
APOC-III	2.10 ± 0.92	1.88 ± 1.03	<0.001	>Q2	304 (53.4%)	162 (44.8%)	0.007
D-dimer	1.84 ± 0.93	2.51 ± 0.84	<0.001	>Q2	217 (38.1%)	247 (69.0%)	<0.001
Endostatin	1.89 ± 0.84	2.49 ± 0.97	<0.001	>Q2	222 (38.0%)	245 (68.2%)	<0.001
GroA	1.97 ± 0.97	2.02 ± 0.97	0.286	---	---	---	
IL-6	1.77 ± 0.93	2.35 ± 0.89	<0.001	>Q2	214 (39.4%)	232(67.1%)	<0.001
NT-proBNP	1.86 ± 0.88	2.59 ± 0.88	<0.001	>Q2	220 (38.6%)	247(68.2%)	<0.001
VAP-1	1.97 ± 0.96	2.21 ± 1.05	<0.001	>Q3	118 (20.7%)	115 (31.9%)	<0.001
vWF	1.87 ± 0.99	2.28 ± 0.89	<0.001	>Q2	246 (42.9%)	224 (61.7%)	<0.001
IGFBP-3	1.97 ± 0.91	2.05 ± 1.09	0.208	---	---	---	
FAS-L	1.99 ± 0.98	1.96 ± 1.00	0.677	---	---	---	
TNF-R1	1.82 ± 0.85	2.45 ± 1.03	<0.001	>Q2	205 (38.3%)	226 (69.5%)	<0.001
NCAM	1.97 ± 0.99	2.03 ± 0.97	0.416	---	---	---	
S100B	1.99 ± 0.99	2.07 ± 0.96	0.322	---	---	---	
Hsc70	2.04 ± 0.94	2.06 ± 0.97	0.757	---	---	---	

APOC-III: apolipoprotein CIII (µg/mL); GroA: growth-related oncogene α (pg/mL); IL-6: interleukin 6 (pg/mL); NT-proBNP: N-terminal pro-B-type natriuretic peptide (pg/mL); VAP-1: vascular adhesion protein-1 (pg/mL); vWF: von Willebrand factor (%); IGFBP-3: insulin-like growth factor binding protein-3 (pg/mL); FasL: Fas ligand (pg/mL); TNF-R1: tumor necrosis factor receptor-1 (pg/mL); NCAM: neural cell adhesion molecule (pg/mL); S100B: S100 calcium-binding protein B (pg/mL); Hsc70: heat shock 70 kDa protein-8 (ng/mL).

**Table 3 life-11-00135-t003:** Blood biomarker mean values and best quartile cut-off points to predict mortality due to stroke during follow-up.

Biomarkers	Death Due to Stroke		*p*-Value	Cut-Off (Quartile)	Death Due to Stroke		*p*-Value
	No (*n* = 804)	Yes (*n* = 133)			No (*n* = 804)	Yes (*n* = 133)	
APOC-III	2.03 ± 0.98	1.95 ± 0.93	0.329	---	---	---	
D-dimer	2.02 ± 0.95	2.58 ± 0.84	<0.001	>Q2	371 (46.4%)	93 (72.1%)	<0.001
Endostatin	2.06 ± 0.91	2.58 ± 1.05	<0.001	>Q3	173 (21.7%)	61 (46.2%)	<0.001
GroA	1.97 ± 0.97	2.11 ± 1.01	0.103	---	---	---	
IL-6	1.94 ± 0.95	2.40 ± 0.89	<0.001	>Q2	361 (47.3%)	85 (68.0%)	<0.001
NT-proBNP	2.07 ± 0.94	2.59 ± 0.92	<0.001	>Q2	382 (47.8%)	85 (63.9%)	<0.001
VAP-1	2.03 ± 0.99	2.24 ± 1.05	0.021	>Q3	190 (23.8%)	43 (32.6%)	0.027
vWF	1.98 ± 0.97	2.35 ± 0.89	<0.001	>Q2	386 (48.0%)	84 (63.2%)	<0.001
IGFBP-3	2.01 ± 0.96	1.97 ± 1.14	0.768	---	---	---	
FAS-L	1.98 ± 0.99	1.98 ± 0.98	0.945	---	---	---	
TNF-R1	1.99 ± 0.94	2.51 ± 1.09	<0.001	>Q2	351 (47.1%)	80 (69.6%)	<0.001
NCAM	1.97 ± 0.99	2.11 ± 0.93	0.177	---	---	---	
S100B	2.01 ± 0.99	2.09 ± 0.90	0.434	---	---	---	
Hsc70	2.04 ± 0.96	2.11 ± 0.88	0.421	---	---	---	

ApoCIII: apolipoprotein CIII (µg/mL); GroA: growth-related oncogene α (pg/mL); IL-6: interleukin 6 (pg/mL); NT-proBNP: N-terminal pro-B-type natriuretic peptide (pg/mL); VAP-1: vascular adhesion protein-1 (pg/mL); vWF: von Willebrand factor (%); IGFBP-3: insulin-like growth factor binding protein-3 (pg/mL); FasL: Fas ligand (pg/mL); TNF-R1: tumor necrosis factor receptor-1 (pg/mL); NCAM: neural cell adhesion molecule (pg/mL); S100B: S100 calcium-binding protein B (pg/mL); Hsc70: heat shock 70 kDa protein-8 (ng/mL).

## Data Availability

The data that support the findings of this study are available on request from the corresponding author, A.B.

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
