# Peer review of "Blood Biomarkers to Predict Long-Term Mortality after Ischemic Stroke"

_life, 2021, doi:10.3390/life11020135_

Round 1
Reviewer 1 Report
This is an interesting study, with clearly described results. Overall it is easy to read and understand.
Author Response
Dear reviewer, thank you for considering our manuscript “Blood biomarkers to predict long term mortality after ischemic stroke” for publication in the special issue Innovative Stroke Diagnostic and Treatment Strategies for Life.
We would like to thank the reviewer for the time and effort to review the manuscript.
As suggested by another reviewer we have sent the manuscript to the English editing service of MDPI (English editing ID: English-26743) to improve the quality of the text.
Reviewer 2 Report
Laura Ramiro and colleagues showed that the acute upregulation of endostatin, IL-6, and 337 TNF-R1 after ischemic stroke can predict long-term mortality. The manuscript is well presented. However, the authors need to revisit the text. At times, texts are not in place and there are typos in the texts.
Author Response
Dear reviewer, thank you for considering a revised version of our manuscript “Blood biomarkers to predict long term mortality after ischemic stroke” for publication in the special issue Innovative Stroke Diagnostic and Treatment Strategies for Life.
We would like to thank the reviewer for the time and effort to review the manuscript.
We are grateful to the Reviewer for pointing out the necessity to improve the writing quality of the manuscript. For that reason, we have sent the manuscript to the English editing service of MDPI (English editing ID: English-26743) to upgrade the quality of the text. We attach the English editing certificate. All the changes have been incorporated to the text and highlighted in red.
We hope that these modifications make the original manuscript more easy to understand.

Reviewer 3 Report
The authors look for mortality blood markers with predictive power after stroke. The patients used were already included in a previous study and the authors did fallow ups for about 5 years.
The article is easy to read. The only suggestion I will make and encourage the authors to implement is to me a bit more clear with the description of the patients used. My biggest concern was that I needed to open the original paper to see what the inclusion and exclusion criteria were for the 941 patients tracked for this paper.
Author Response
Dear reviewer, thank you for considering a revised version of our manuscript “Blood biomarkers to predict long term mortality after ischemic stroke” for publication in the special issue Innovative Stroke Diagnostic and Treatment Strategies for Life.
We would like to thank the reviewer for the time and effort to review the manuscript.
We are grateful to the Reviewer for pointing out the necessity to improve the description of the patients used. For that reason, the original manuscript has been modified considering the reviewer concern, incorporating the inclusion and exclusion criteria of the patients (line 68-73):
In summary, inclusion criteria were suspected stroke at first medical assessment with persisting symptoms when arriving at the emergency room; age >18 years; <6 hours from symptom onset to blood sample collection; blood collection preceding thrombolytic treatment and signed informed consent. The only exclusion criteria was impossibility to collect blood samples. Moreover, patients with an unclear diagnosis 1 month after the index event were excluded from the analysis.
We hope that this modification makes the original manuscript more clear to understand.
In addition, as suggested by another reviewer we have sent the manuscript to the English editing service of MDPI (English editing ID: English-26743) to improve the quality of the text.